# Emerging Arboviruses of Public Health Concern in Africa: Priorities for Future Research and Control Strategies

Yusuf Amuda Tajudeen [1,2,*], Habeebullah Jayeola Oladipo [1,3], Iyiola Olatunji Oladunjoye [1], Rashidat Onyinoyi Yusuf [3], Hammed Sodiq [1], Abass Olawale Omotosho [4], Damilola Samuel Adesuyi [5], Sodiq Inaolaji Yusuff [6] and Mona Said El-Sherbini [7,*]

[1] Department of Microbiology, Faculty of Life Sciences, University of Ilorin, P.M.B. 1515, Ilorin 240003, Nigeria
[2] Department of Epidemiology and Medical Statistics, Faculty of Public Health, College of Medicine, University of Ibadan, P.M.B. 5017 G.P.O., Ibadan 200212, Nigeria
[3] Faculty of Pharmaceutical Sciences, University of Ilorin, P.M.B. 1515, Ilorin 240003, Nigeria
[4] Department of Microbiology, Faculty of Pure and Applied Sciences, Kwara State University, P.M.B. 1530, Malete-Ilorin 23431, Nigeria
[5] Department of Microbiology, Faculty of Science, Adekunle Ajasin University, P.M.B. 001, Akungba-Akoko 342111, Nigeria
[6] Department of Medicine, Faculty of Clinical Sciences, Obafemi Awolowo University, P.M.B. 5538, Ile-Ife 220101, Nigeria
[7] Department of Medical Parasitology, Faculty of Medicine, Cairo University, Cairo 11562, Egypt
* Correspondence: tajudeenamudayusuf@gmail.com (Y.A.T.); monas.elsherbini@kasralainy.edu.eg (M.S.E.-S.); Tel.: +234-(0)-706-206-3691 (Y.A.T.); +20-100-246-5704 (M.S.E.-S.)

**Abstract:** Arboviruses are most prevalent in tropical and subtropical regions, where arthropods are widespread. The World Health Organization (WHO) estimated that the mortality burden of arbovirus diseases, such as yellow fever in Africa, was 84,000–170,000 severe cases and 29,000–60,000 deaths in 2013. These epidemics emphasize the urgent need for integrated control and prevention of arboviral diseases. Challenges in managing and controlling arboviral diseases in Africa are mainly attributed to poor insect vector control, insecticide resistance, and poor sanitation and solid waste management. The removal or reduction of mosquito populations amongst susceptible individuals is identified as the most effective measure to control many vector-borne diseases. Current public health needs call for efficient vector control programs and maintenance of adequate surveillance systems through the availability of trained personnel and rapid diagnostic facilities, providing an interdisciplinary response to control and mitigate the threats of emerging and re-emerging arboviruses. Furthermore, research priorities should focus on understanding the factors responsible for adaptation to other vectors, determinants of infection and transmission, and the development of high efficiency antiviral molecules or candidate vaccines. Here, we explore and review our current understanding of arboviruses of public health importance in Africa, with a focus on emerging arboviruses, their arthropod vectors, and the epidemiology of major arboviruses. Finally, we appraise the role of planetary health in addressing the threat of arboviruses and identify other priority areas of research for effective control.

**Keywords:** arboviruses; arthropods; epidemics; Africa; planetary health

## 1. Introduction

Globally, arboviruses are one of the persistent causes of human and animal diseases, infecting millions of individuals and imposing significant social and economic burdens [1]. Approximately 73% of current emerging and re-emerging pathogenic agents have arboviral origins, and approximately 60% of the 1500 or more infectious microorganisms that are known human pathogens are recognized as zoonotic [2]. Arboviruses are generally transmitted by arthropod vectors, such as mosquitoes, ticks, sandflies, and midges, along with other hematophagous arthropods [1,3–6]. *Aedes* mosquitoes are the most important

arboviral vectors; the two main species of which, *Ae. aegypti* and *Ae. albopictus*, allow the transmission of medically crucial viruses such as Chikungunya virus (CHIKV), Dengue virus (DENV), and Yellow fever virus (YFV) [5]. Most of these arboviruses belong to four virus families: Togaviridae (genus Alphavirus), Flaviviridae (genus Flavivirus), Bunyaviridae (genera Orthobunyavirus, Phlebovirus, and Nairovirus), and Reoviridae (genera Coltivirus and Orbivirus) [5,7]. Human or animal infections can range from subclinical-mild to encephalic or hemorrhagic with high fatality rates; however, arthropods infected with arboviruses do not show detectable signs of infection, even though the arthropod can harbor the virus for life [1,8]. This can partially be explained by the evolutionary biological adaptations developed by arboviruses inside their arthropod vectors over the years, thereby presenting a sort of symbiotic relationship on the one hand, and preventing morbidity and mortality in arthropod vectors on the other hand, thus maintaining the propagation of arboviruses in the environment [9].

The last few decades have witnessed dramatic epidemics of emerging and re-emerging arboviral diseases, posing serious global public health risks [1,3,10]. The most common risk factor of DENV is the co-circulation of multiple serotypes (hyperendemic), which is associated with the emergence of the severe form of the disease, dengue hemorrhagic fever/shock syndrome (DHF/DSS) [1]. The propagation of DENV is based on biological transmission by mosquito vectors living in close association with humans, that is, relying on the human host as the reservoir and implicating the host. CHIKV was recognized as the etiologic cause of febrile disease epidemics in the 1950s and continues to be an important pathogen in Southeast Asia in a strictly peridomestic mosquito cycle, unlike in Africa, where there is evidence of a sylvatic cycle involving arboreal mosquitoes and nonhuman primates. However, there was a re-emergence in 2005 in East Africa and the Southern Indian Ocean Islands that led to a succession of massive outbreaks [10].

Arboviruses associated with human and animal diseases are most prevalent in tropical and subtropical regions, where arthropods are widespread; nonetheless, many arboviruses circulate among wildlife species in temperate regions of the world [1]. Excluding the global distribution of viruses such as West Nile virus (WNV), DENV, and CHIKV, the majority of arboviruses are endemic but limited to specific regions of the world, although dispersal to distant locations may occur through vector mitigation [8].

Furthermore, there will be significant concentrations of susceptible human hosts when humans are exposed to arboviruses through anthropogenic activities impacting global warming and deforestation associated with urbanization. Climate change will have a significant impact on human and animal movement as a result of changes in land use and housing designs—which can further add to the complexity of arboviral emergence [11]. Contact between humans and vectors has increased owing to deforestation associated with urbanization. Urban expansion has led to high concentrations of susceptible human hosts living in socioeconomic conditions favorable to the expansion of vector populations. This facilitates viral transmission and outbreaks of epidemics [12]. These examples generally enhance human-vector contact, promoting viral transmission and epidemic outbreaks. The greatest risks faced by humans are due to the ability of some arboviruses to adopt urban transmission cycles that involve highly efficient and anthropophilic vectors, such as *Ae. aegypti* and *Ae. albopictus*, or peridomestic enzootic cycles [12].

The recent emergence and transmission of the Chikungunya virus in East Africa, as well as the Dengue virus in some parts of the tropics and subtropics, have re-emphasized the need for immediate action against emerging and re-emerging arboviruses [10,13,14]. These epidemics emphasize the urgency and dire need for integrated control and prevention of arboviral diseases, especially those transmitted by *Aedes* mosquitoes in urban areas [10,15,16]. Moreover, the re-emergence of Yellow fever virus has been continuously reported in tropical countries such as Brazil, despite vaccine the availability and administration of vaccines [17,18]. Furthermore, prevention and control strategies focused on vector control, which involves the use of insecticides, environmental management, and social mobilization, have not been effective in practice [10]. It is evident that viral tropism in

different hosts and arthropod species indicates that no single strategy can fully handle the issue of arbovirus emergence and re-emergence. For instance, the use of insecticides and long-lasting treated nets has its limitations in that arboviruses have developed resistance to insecticides and pyrethroids used in long-lasting treated nets (LLTNs). Understanding the various factors that contribute to the (re)emergence of arboviruses will help to address these challenges, coupled with a holistic approach.

In this review, we discuss our current understanding of arboviruses of public health significance in Africa. The article is divided into five major sections. In Section 1, which is the introduction, we provide an overview of arboviruses in Africa. Section 2 deals with emerging arboviruses and their arthropod vectors. In this section, we highlight various vectors of arboviruses and the socio-ecological and biological risk factors contributing to their prevalence and infectivity rate. In Section 3 of the article, we explore the epidemiology of major arboviruses in Africa, including Dengue, Zika, Chikungunya, and Yellow fever viruses, coupled with ongoing management and control strategies to bring them under control and the challenges involved. The second objective of the article is to examine the potential role of a planetary health approach in addressing the public health threat of arboviruses and we suggest priority research areas to achieve effective control, which are discussed in Sections 4 and 5, respectively.

## 2. Emerging Arboviruses in Africa and Their Arthropod Vectors

Arboviruses are known to have a variety of arthropod species capable of infecting the predominant host species, such as mosquitoes and ticks, in addition to different mechanisms of disease emergence [19]. More than 300 mosquito species can transmit arboviruses, among which *Aedes* and *Culex* are the most common species associated with the transmission of arboviruses [20]. Approximately 116 different species of tick are also vectors that transmit arboviruses. Additionally, 25 midge species have been shown to transmit arboviruses, including *Culicoides* (24 subspecies) and *Lasiohelea*. Other vectors, such as mites, lice, sandflies, bed bugs, stinkbugs, gadflies, and blackflies, have also been reported to transmit arboviruses [12]. This diversity of species, along with the wide distribution of vectors, explains why arboviruses can spread successfully worldwide and particularly in temperate regions such as Africa, where arthropod breeding conditions are favorable. Table 1 highlights various emerging arboviruses in Africa, their distribution, hosts, and arthropod vectors [21].

**Table 1.** Emerging Arboviruses in Africa and their Arthropod Vectors [21].

| Arboviruses | Families | Disease (s) | Vector | Vertebrate Host | Distributions |
|---|---|---|---|---|---|
| Ilesha Virus (ILEV) | Bunyaviridae | Ilesha viral disease | *Anopheles gambiae* | Humans | Cameroon, Ghana, Niger, Nigeria, Senegal, Uganda, Madagascar |
| Bouboui (BOUV) | Flaviridae | Boubouii disease | *Aedes africanus* | Unknown | Central Africa |
| Dengue Virus (DENV1) | Flaviridae | Dengue fever | *Aedes aegypti* | Human and Non-human primates | Angola, Benin, Burkina Faso, Cameroon, Cape Verde, Nigeria, Uganda, Zambia, and Togo. |
| Uganda S (UGSV) | Flaviridae | Uganda S disease | *Aedes longipalpis* | unknown | Uganda |
| West Nile Virus (WNV) | Flaviridae | West Nile Virus disease (Can event to Encephalitis or Meningitis) | Mosquito | Birds, horse | West Nile District of Uganda |
| Yellow Fever Virus (YFV) | Flaviridae | Yellow Fever | *Aedes aegypti, A. africanus.* | Monkeys, Chimpazees, Baboons, Humans | Angola, Benin, Burkina Faso, Cameroon, Central African Republic, Ethiopia, Gabon, Gambia, Ghana, Kenya, Liberia, Mali, Nigeria |
| Zika Virus (ZIKV) | Flaviridae | Zika Fever or Zika Virus Disease | *Aedes aegypti* | Monkeys | Zika forest in Uganga, Tanzania |
| Chikungunya Virus (CHIKV) | Flaviridae | Chikungunya Fever | *Aedes* Mosquitoes especially *Aedes aegypti* and *Aedes albbopictus* | Human | East/Central/South Africa (ECSA) |
| Rift Valley Fever Virus (RVFV) | Flaviridae | Rift Valley Fever | *Aedes aegypti, A. cumminsii* | Cattle, Sheep, Goats, Humans | Angola, Botswana, Burkina Faso, Cameroon, Chad, Congo, Egypt, Gabon, Gambia, Guinea, Namibia, Niger, Nigeria, Senegal, South Africa, South Sudan, Tanzania, Uganda |
| Mossuril Virus (MOSV) | Rhabdoviridae | Mossuril Viral disease | *Culex sp.* | unknown | Mozambique, South Africa |
| African Horse Sickness Virus (AHSV) | Reoviridae | African Horse Sickness | Culicoides species; *Culicoides imicola* and C. obsoletus | Horse | Southern Africa |

*Risk Factors Contributing to the Emergence of Arboviruses*

The trilogy of viral pathogen, mosquito/tick vector, and susceptible virus-replicating host (humans and animals) are entities necessary for the transmission of arboviral diseases. Therefore, the assessment of possible risk factors associated with these entities is essential for a comprehensive assessment of arboviral prevalence. The risk factors highlighted in several studies include the human population, viral mutation, vector adaptation and diversity, climate change, and anthropogenic activities [21–24].

In 2012, the global human population reached 7 billion people and it is estimated to reach 9.6 billion by 2050 [25]. This increase in population size has the propensity to drive urbanization, which is an important factor that can facilitate the migration of vectors and accessibility to host species, and consequently, the disease transmission rate will increase due to frequent human interactions with vectors from habitats previously destroyed for settlement or industrialization. Additionally, the increasing population will result in a corresponding increase in the disposal of waste products in the environment, resulting in more breeding sites for vectors such as *Aedes* and *Culex* [21].

Another highlighted risk factor is the ability of the causative virus to mutate. These mutations (minor or major) could give rise to pathogens with increased virulence, transmissibility, and pathogenicity. Such mutations are facilitated by the simultaneous infection of mosquitoes with two viruses, possibly due to interrupted feeding or infected intermittent blood meals [21]. For example, a mutation of the Chikungunya virus envelope glycoprotein E1 may have led to enhanced viral replication and subsequent transmission by *Ae. albopictus*, resulting in the spread of the disease to Asian regions [26,27]. Similar to viruses, disease vectors are also capable of mutations to better adapt to their environment. A typical example is *Ae. aegypti*, which was historically a forest-dwelling species. This vector is presently known to be highly anthropophilic and well- adapted to urban environments. Consequently, diseases transmitted by this vector have spread to several tropical and subtropical parts of the world [21].

It is important to note that most arboviral vectors have short life spans. For example, the *Ae. aegypti* mosquito has four developmental stages, which include the egg, larva, pupa, and adult. However, this process takes approximately 8–10 days, depending on factors such as larval density, food availability, and temperature [28]. The reproductive capacity of female mosquitoes only requires a single sperm dose to produce a batch of eggs. Under favorable conditions, the female may lay approximately 100 eggs at a time in a capful of water, enough for the eggs to lay and hatch into the adult stage within one week [1]. The high durability and adaptability of the aquatic egg stage facilitate its survival in desiccated areas for long periods of more than one year [29]. While male mosquitoes are not blood feeders, female *Ae. aegypti* have a strong preference for human blood, while *Ae. albopictus* are less discriminate and can feed on both animals and humans [30]. An estimated 3–4 μL blood can be ingested by *Ae. aegypti* and the vector may become infected by arboviruses after feeding on (viremic) human blood, infecting the mosquito throughout its lifespan. In a single feeding episode, female *Ae. aegypti* have the ability to transmit more than one arbovirus and consume multiple blood meals to complete their gonotrophic cycle, which usually takes 2–8 days from the blood meals to the egg laying process [30]. Female *Ae. aegypti* rarely migrate beyond 30–40 m from households where they develop as larvae and grow into the adult stage for urban transmission; however, they can disperse 40–80 m for the rest of their lifetime [31,32]. Flying to new areas has been shown to increase the ability of invasive *Ae. aegypti* to replace the resident mosquito population through competitive exclusion, and this can improve infectivity rates and propagation of arbovirus infection [33].

Furthermore, climate change has been observed to influence the emergence of arboviral diseases, and this can be attributed to the effects of climate change on the distribution (geographic and temporal) and life cycles of vectors, pathogen dispersal patterns, viral mutations, and viral transmission. Such variations can lead to the emergence of arboviral diseases [24,34]. Anthropogenic activities, such as trade, tourism, and migration, have also

been implicated in several arboviral disease outbreaks. A classic example is Yellow fever virus, which was reportedly transmitted from Africa to other parts of the world during slave trading in the 1650s [24,35].

## 3. Epidemiology of Major Arboviruses in Africa

Escalating global warming, population mobility due to wars or urbanization, insecticide resistance, and increased vector breeding sites due to poor sanitary conditions have contributed immensely to the emergence and re-emergence of Dengue, Zika, Chikungunya, and Yellow fever arboviral infections throughout different geographical locations, especially in tropical, subtropical, and developing African countries [36,37]. This is discussed in detail in the following sections.

### 3.1. Dengue Virus

In Africa, epidemics of Dengue virus (DENV) were documented in Zanzibar (1823, 1870), Burkina Faso (1925), Egypt (1887, 1927), South Africa (1926–1927), and Senegal (1927–1928) in the late 19th and early 20th centuries [38,39]. The first epidemics in the Americas were documented in the French West Indies in 1635 and Panama in 1699, both of which were likely caused by DENV [40]. Between 60 and 140 million clinically evident cases of dengue fever/hemorrhagic fever/shock syndrome have been reported each year due to the virus [14]. Dengue fever affects approximately 400 million people worldwide each year, resulting in 22,000 deaths [41]. According to the WHO estimate, approximately 1.6 million dengue cases were reported in North and South America in 2010, with 49,000 severe cases.

### 3.2. Zika Virus

Zika virus is spread by non-human primates and sylvatic mosquitoes such as *Aedes africanus* [42]. As the number of cases of congenital Zika virus increased in babies born in Brazil, the World Health Organization in 2016 declared Zika (ZIKV) a 'Public Health Emergency of International Concern' [9]. The discovery of neutralizing antibodies in human serum from East Africa in 1952 provided the first indication of human infection. In 2008, a ZIKV outbreak erupted on Yap Island in the Federated States of Micronesia. ZIKV is still a threat to human health around the world today [37]. ZIKV has been reported in 26 African nations.

### 3.3. Chikungunya Virus

Chikungunya fever virus (CHIKV) resurfaced on the east coast of Africa in 2004–2005, as well as on the east African islands of Lamu and Madagascar [37]. Compared to DENV and ZIKV, CHIKV is believed to generate a higher rate of symptomatic infections [31]. The virus is common in rural regions and results in sporadic infections. The virus has resulted in major outbreaks in metropolitan areas, infecting a large percentage of the population in a matter of weeks. Multiple outbreaks of CHIKV have been reported throughout Africa [31]. In the period from the first CHIKV outbreak in Tanzania in 1952–1953 to today, evidence of CHIKV infection and serological evidence of previous exposure have been reported in 33 African countries [37]. Some of these countries include Angola, Benin, Burundi, Cameroon, Central African Republic, Cote d'Ivoire, the Democratic Republic of the Congo, Djibouti, Guinea, Kenya, Madagascar, Malawi, Mauritius, Mayotte, Mozambique, and Nigeria.

### 3.4. Yellow Fever Virus

Sporadic cases of Yellow fever virus (YFV) and brief epidemics have been reported in West Africa since the end of the 15th century [37]. The disease was non-existent in East and South Africa at that time due to the lack of an adequate vector [31]. YFV outbreaks were also documented in Central and West Africa during the 1960s and 1980s [43]. YFV epidemics were reported in Ethiopia between 1960 and 1962, resulting in 100,000 cases and 30,000 deaths [44]. A YFV outbreak was also reported in Nigeria between 1984 and

1990, resulting in 21,299 deaths [45]. As recently documented in Angola and the Republic of Congo, yellow fever epidemics can quickly spread in highly populated metropolitan areas [12]. The mortality rate of YFV has been estimated to be approximately 20–60% [46].

### 3.5. Ongoing Management and Control Strategies for Arboviral Disease in Africa

The challenges in managing and controlling arboviral diseases are mainly attributed to poor insect vector control, insecticide resistance, poor sanitation and solid waste management, as well as inadequate household water supply and disposal. Prevention is always considered better than cure, and the main control method for arboviruses has relied on chemical insecticides such as pyrethroids, organochloride, and organophosphorus, which primarily act as neurotoxins affecting the vector's nervous system [47]. The use of a biological agent, such as the bacteria *Bacillus thuringiensis* subsp. *israelensis* (Bti) isolated in 1976, was found to be toxic to mosquito larvae [48]. *Bacillus sphaericus* has also been found to have a similar potency against mosquitoes [49]. Numerous biological measures, such as the use of fungi, plants, and fish, have also been employed to control the growth and propagation of the mosquito population [49]. Reduction of breeding sites and limiting vector-host contact (that is, the use of barrier protection, such as bed nets, removal of all objects, storage of unwanted water collection, waste disposal on the streets, and proper maintenance of sewage systems and canals) have been in used as control strategies for ages [50]. There is no vaccine alternative for the vast majority of arboviruses; exceptions to this include the 17D Yellow fever virus (YFV) vaccine and the newly approved Dengue virus vaccine (DENV), Dengvaxia [51]. Therefore, additional and alternative control strategies need to be developed in Africa and other countries due to increasing resistance to insecticides, coupled with the toxic effects of organophosphorus compounds on human and animal health, and the risk of sporadic spread of infectious arthropods to new areas [52]. Transmission blocking vaccines (TBVs) are attractive tools currently used to decrease arboviral transmission, especially in the absence of specific antiviral treatments to prevent severe infection in high-risk populations, such as the elderly and pregnant women [53]. TBV works by targeting the transmission capacity of the vector, which prevents the pathogen from completing its life cycle in the arthropod vector, and thus halts transmission to human hosts [54]. According to a study conducted by Londono-Renteria and colleagues in 2016 [53], a vaccine formulation that combined both pathogen and arthropod key target molecules could increase the efficiency of TBVs. However, care must be taken to avoid cross-reactivity or autoimmune diseases in humans. TBVs have been found to have the potential to decrease infection among certain populations while increasing herd immunity, making them an attractive tool to combat pathogen transmission and proliferation. Thus, more research is needed for the development of vaccines that are safe and affordable.

### 3.6. Challenges in Addressing the Risk of Arboviruses in Africa

Although several interventions, such as vector control via insecticides and vaccine use (17D vaccine and Dengyaxia), have been applied to control arboviral diseases in Africa, continued disease incidence and outbreaks expose the challenges in implementing these interventions [55]. One of these challenges is the increasing resistance of mosquitoes (*Aedes*) to insecticides such as DDT (dichlorodiphenyltrichloroethane), organophosphate adulticides, pyrethroids, and carbamates in several African countries, including Cameroon, Central African Republic, Gabon, Ghana, Nigeria, Senegal, Sudan, and Tanzania, thus limiting their long-term use and efficiency in Africa [55,56]. Accompanying this challenge is a lack of surveillance data on vector species populations in Africa, which makes it difficult to identify regions in need of swift intervention regarding vector population control [57].

Another challenge is the production of vaccines and their use. Although several resources have been invested in vaccine development for arboviral diseases, most viruses (including Zika and Chikungunya) are still without an approved vaccine for effective prevention. Moreover, the public's disposition towards approved vaccines is not favorable for effective control, which is largely facilitated by public concerns about the side effects and

efficacy of these vaccines [8,10]. Furthermore, the absence of an integrated and functional surveillance system for arboviral diseases in most African countries has contributed to the difficulties in accurately tracking the prevalence of these diseases. This challenge, coupled with several cases of misdiagnosis (attributed to the similarity of disease symptoms with other common infectious diseases such as malaria), has affected proper treatment and hence, appropriate measures for outbreak prevention, especially in resource-limited African countries such as Cameroon, Kenya, and Tanzania [55].

Moreover, some African countries, such as Sudan, Kenya, and Uganda, are plagued by limited diagnostic capacity. These countries lack highly advanced sensitive diagnostics capable of early detection of the disease, which has affected the response to disease outbreaks [37,58–60]. Another major challenge is the lack of awareness of arboviral diseases amongst the general populace, especially livestock farmers, which is quite common in several African countries, including Kenya, South Africa, and Nigeria. This health promotion deficit has resulted in poor knowledge, attitude, and practices amongst the general populace in affected countries. In addition, the lack of public trust in the appropriate treatment options offered by healthcare services has contributed to the spread of arboviral diseases in Africa [61–64].

## 4. Planetary Health and Arboviruses

The increased circulation of arboviruses driven by anthropogenic activities such as urbanization, land use change, and deforestation, which have led to climate change and loss of biodiversity, provides a favorable avenue for arthropod vectors and their viruses to thrive and can only be effectively addressed through a planetary health approach [65,66]. As defined by the Rockefeller Foundation-Lancet Commission on Planetary Health, "*Planetary Health is the achievement of the highest attainable standard of health, wellbeing, and equity worldwide through judicious attention to the human systems—political, economic, and social—that shape the future of humanity and Earth's natural systems that define the safe environmental limits within which humanity can flourish*". The flourishing of humanity and planet Earth will require dealing with inter-related threats to planetary health, including arbovirus infections [67]. Climate change is an important factor that facilitates the geographical distribution of vectors and the viruses they carry, including the susceptibility of reservoir hosts to viral infection [65,66]. Properly understanding host-pathogen interactions in an ever-changing climate requires a focus on the Stockholm Paradigm—which is a newly emerging concept viewed under the planetary health lens. The Stockholm Paradigm addresses the climate-related health threat of infectious diseases, leveraged on ecological fittings, the geographical mosaic of co-evolution, taxon pulses, and the oscillation hypothesis. The application of both a planetary health approach and the Stockholm paradigm in tackling the profound threat of arboviral diseases in Africa can be achieved through the implementation of well-established and coordinated entomological surveillance coupled with adequate monitoring of arthropod vectors to allow for early detection and response [68]. Since the planetary health approach advocates for intersectoral collaboration and global partnership, this will allow for cost-resource and knowledge sharing across related translational fields within the concerned stakeholders in response to the arboviral threat at human and environmental interfaces.

## 5. Conclusions

Presently, arboviruses have a well-known history of emergence and a tendency to reoccur in the future. Many unidentified arboviruses exist and high mutation rates contribute to the concern that emerging epidemic strains might be anticipated in the coming years. The world is in dire need of a continuous international and interdisciplinary response to improve the ability to anticipate, control, and mitigate the threats of emerging and re-emerging arboviruses.

Research priorities should be placed on surveillance systems and understanding the factors responsible for vectorial competency, determinants of host infection, and trans-

mission cycles, as well as the development of antiviral molecules or candidate vaccines. Similarities in the characteristics of these viruses could stimulate common research themes for the development of antiviral therapies and vaccines. Additionally, evaluating the available and developing vector control tools is needed to identify the most effective techniques to use in conjunction with these vaccines.

Presently, the best approaches for controlling the majority of vector-borne diseases rely on reducing human-vector contact, and the most effective and efficient method to achieve this goal remains the removal or reduction of mosquito populations amongst susceptible individuals. The implementation of localized arthropod control measures during epidemics in urban and rural areas can play an important role in reducing the impact of arboviruses on humans and animals, provided that these efforts are supported by re-assessed and improved surveillance systems. In addition, the socioeconomic and environmental factors driving the proliferation of vectors, particularly in rural and under-served communities, should be mitigated through integrated research programs and educational awareness programs.

Furthermore, it is important to identify and understand viral genetics, antigenic properties, virulence patterns, vector associations, and maintenance mechanisms in order to control future arboviral outbreaks. Current public health needs include better communication about vector-borne diseases to the population and physicians, guarantees of vector control programs, and maintenance of adequate surveillance systems by ensuring the availability of trained personnel, rapid diagnostic facilities, and appropriate therapeutics.

Finally, adopting a holistic approach, such as 'planetary health,' that encompasses multisystem and multilateral strategies aimed at involving governments, academics, humans, animals, and environmental health partners at national and global levels, as well as international health organizations, could improve knowledge translation between local communities and strengthen control and research programs on arboviral infections through multidisciplinary efforts, while maintaining the ecological balance of our environment.

**Author Contributions:** Conceptualization, Y.A.T., H.J.O. and I.O.O.; methodology, Y.A.T., H.J.O., I.O.O., R.O.Y., H.S., A.O.O., D.S.A., S.I.Y. and M.S.E.-S.; resources, Y.A.T., H.J.O., I.O.O., R.O.Y., A.O.O., H.S. and D.S.A.; data curation, Y.A.T., H.J.O., I.O.O., R.O.Y., H.S., A.O.O., D.S.A., S.I.Y. and M.S.E.-S.; writing—original draft preparation, Y.A.T., H.J.O., I.O.O., R.O.Y., H.S., A.O.O., D.S.A., S.I.Y. and M.S.E.-S.; writing—review and editing, Y.A.T., H.J.O., I.O.O., R.O.Y., H.S., A.O.O., D.S.A., S.I.Y. and M.S.E.-S.; supervision, M.S.E.-S., Y.A.T., H.J.O. and I.O.O. All authors have read and agreed to the published version of the manuscript.

**Funding:** This research received no external funding.

**Institutional Review Board Statement:** Not applicable.

**Informed Consent Statement:** Not applicable.

**Data Availability Statement:** Not applicable.

**Acknowledgments:** We express our gratitude to the reviewers and editors of this journal for their constructive comments and suggestions that enrich the quality of our manuscript.

**Conflicts of Interest:** The authors declare no conflict of interest.

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
