# Peer review of "Emerging Arboviruses of Public Health Concern in Africa: Priorities for Future Research and Control Strategies"

_challenges, doi:10.3390/challe13020060_

Round 1

Reviewer 1 Report

Review

Overview: According to the WHO, arboviruses pose a major health risk, especially in tropical and subtropical regions. This shows the urgency and urgent need for integrated control and prevention of arboviral diseases. The article (viewpoint) outlines the challenges in preventing and resolving these issues and priorities for research.

Evaluation: The manuscript is a viewpoint, it raises a very important issue. The manuscript is scientifically sound, clearly structured and comprehensive. The table (Supplementary Material) is concise and clear and properly shows the data. The cited references are appropriate. This article makes an important contribution to further research and development.

Author Response

Dear Reviewer,

Thank you for taking the time to review our manuscript, your insightful suggestions and comments have, indeed, strengthen our manuscript. We also appreciate your kind words towards our manuscript.

Thank you.

Sincerely,

Authors

Reviewer 2 Report

In recent decades, we have observed the emergence and re-emergence of arboviral diseases. These diseases indicate serious risks and future challenges for global public health. The authors provide an accurate and detailed review of arbovirus infections in Africa.

Author Response

(The authors gave the same response as above.)

Reviewer 3 Report

This is certainly an interesting topic, the authors have addressed. But I am concerned that the manuscript has several parts that must need improvement. Here are some of my comments, but these are only suggestive examples, not all of them. Please find similar contexts all over the manuscript and revise the manuscript carefully.

1. Line 63: Three articles have been cited in support of the statement "dramatic emerging and re-remerging epidemics of arboviral diseases....". I would suggest to provide a gist of the epidemics reported in the three articles.

2. Line 64-65 is repeated. It is already in the abstract.

3. Lines 70-72: the statement is very crucial and of great concern. I wish to see an elaborative focus on this issue regarding global worming and deforestation with respective references. The following lines are concentrated on the urbanization, but missing global worming and deforestation.

4. The last paragraph of the Introduction is hard to follow. I would recommend a rewrite of that. Overall focus would remain same, but the flow of reading is not very lucid.

5. In the section 2.1, I would like to highlight two more aspects: (i) most of the arbovirus vectors have short lifetimes and they cannot migrate very large distance during their lifetime; e.g., mosquito habitats are approximately confined within 50 meters. (ii) there is a bottleneck in the transmission cycle of arboviruses that a single virus particle is sufficient to infect a human/host but to infect a mosquito/vector, one need a minimum MID50(half-maximal mosquito infection dose). Hence, it could be a very efficient way of preventing epidemic spread or reduce the burst-size by reducing the human-to-mosquito transmission. 

6. I expected a section of viewpoint after section 3. The authors have been still reporting cases and incidents in different countries and for different arboviruses up to section 3. But a strong viewpoint or opinion is expected at the end of all these report and discussions. I am afraid the authors failed to propose any viewpoint or opinion at this point of the article. I would strongly recommend to include another section on the opinion.

Author Response

Dear Reviewer,

Thank you for taking the time to review our manuscript, your insightful suggestions and comments have, indeed, strengthen our manuscript. Here below, is how we have addressed your comments and suggestions point-by-point.

Comment:

Line 63: Three articles have been cited in support of the statement "dramatic emerging and re-remerging epidemics of arboviral diseases....". I would suggest to provide a gist of the epidemics reported in the three articles.

Response:

This has been addressed in lines 99-111 of the revised manuscript according to your suggestion.

Comment:

Line 64-65 is repeated. It is already in the abstract.

Response:

This has been addressed in the revised manuscript.

Comment:

Lines 70-72: the statement is very crucial and of great concern. I wish to see an elaborative focus on this issue regarding global warming and deforestation with respective references. The following lines are concentrated on the urbanization, but missing global warming and deforestation.

Response:

This has been addressed in lines 83 -89

Comment:

The last paragraph of the Introduction is hard to follow. I would recommend a rewrite of that. Overall focus would remain same, but the flow of reading is not very lucid.

Response:

The Paragraph has been rephrased based on your suggestion.

Comment:

In the section 2.1, I would like to highlight two more aspects: (i) most of the arbovirus vectors have short lifetimes and they cannot migrate very large distance during their lifetime; e.g., mosquito habitats are approximately confined within 50 meters. (ii) there is a bottleneck in the transmission cycle of arboviruses that a single virus particle is sufficient to infect a human/host but to infect a mosquito/vector, one need a minimum MID50(half-maximal mosquito infection dose). Hence, it could be a very efficient way of preventing epidemic spread or reduce the burst-size by reducing the human-to-mosquito transmission.

Response:

This has been addressed in lines 154-174 of the revised manuscript.

Comment:

I expected a section of viewpoint after section 3. The authors have been still reporting cases and incidents in different countries and for different arboviruses up to section 3. But a strong viewpoint or opinion is expected at the end of all these report and discussions. I am afraid the authors failed to propose any viewpoint or opinion at this point of the article. I would strongly recommend to include another section on the opinion.

Response:

A section on “Planetary Health and Arboviruses Control” has been provided in lines 301-326 of the revised manuscript.

Reviewer 4 Report

This article summarizes the current knowledge and some possible control measures against arbovirus infections in Africa. It specifically addresses the situation in the affected countries in Africa, but from the virological / entomological side it stays mostly superficial. Therefore the relevance of this article for a general audience is somewhat limited.

Some specific comments:

ll 55-57: the wording here may be revised, the authors state that arthopods are often without clinical signs after virus infection, but how can we measure clinical signs of infection in an arthropod? It is also stated that these vectors may harbor the virus over years. This is in my view misleading, since an individual vector will not live for years, unless the different developmental stages are all counted, which should then be made clear to avoid misunderstanding.  

l 91: explain LLTN

l 107: delete ‘spread’ once in this sentence

l 108: delete one . and delete the comma after Table 1.

l 120: from the fact that the global human population is expected to rise in the next years, it is concluded that ‘the host species will be more vulnerable’, which is not necessarily a direct conclusion, which therefore needs some additional explanation.

l 170-171: This sentence ‘Chikungunya fever was first detected on the East coast of Africa in 2004–2005, as well 171 as in the East African islands of Lamu and Madagascar’ is somewhat misleading, as it may implicate that this was the first description of the virus ever, which is not the case (this was already in the 1950ies). Therefore this should be reworded.

ll 216-217: the concept of a transmission blocking vaccine (TBV) is explained as ‘TBV works by halting new infections in transmission-216 competent insects, by focusing on specific arthropod proteins’, which would benefit from a practical example (i.e. a specific protein that is blocked in one of the TBV approaches).

 Supplementary table: this is somewhat incomplete, it would be good to show the respective virus families, and then also order the agents according to the virus family (instead of alphabetically). How were the references selected that are given here?

Author Response

Comments and Suggestions for Authors

This article summarizes the current knowledge and some possible control measures against arbovirus infections in Africa. It specifically addresses the situation in the affected countries in Africa, but from the virological / entomological side it stays mostly superficial. Therefore, the relevance of this article for a general audience is somewhat limited.

Dear Reviewer,

Thank you for taking the time to review our manuscript, your insightful suggestions and comments have, indeed, strengthen our manuscript. Here below, is how we have addressed your comments and suggestions point-by-point.

Some specific comments:

Comments:

55-57: the wording here may be revised, the authors state that arthropods are often without clinical signs after virus infection, but how can we measure clinical signs of infection in an arthropod? It is also stated that these vectors may harbor the virus over years. This is in my view misleading, since an individual vector will not live for years, unless the different developmental stages are all counted, which should then be made clear to avoid misunderstanding. 

Response:

This has been addressed in lines 54-57 in the revised manuscript.

Comments:

91: explain LLTN

Response:

This has been explained in line 107.

Comment:

107: delete ‘spread’ once in this sentence

Response:

This has been addressed in line 94.

Comment:

108: delete one. and delete the comma after Table 1.

Response:

This has been addressed in the revised table 1.

Comment:

120: from the fact that the global human population is expected to rise in the next years, it is concluded that ‘the host species will be more vulnerable’, which is not necessarily a direct conclusion, which therefore needs some additional explanation.

Response:

This has been addressed in the revised manuscript.

Comment:

170-171: This sentence ‘Chikungunya fever was first detected on the East coast of Africa in 2004–2005, as well 171 as in the East African islands of Lamu and Madagascar’ is somewhat misleading, as it may implicate that this was the first description of the virus ever, which is not the case (this was already in the 1950ies). Therefore, this should be reworded.

Response:

This has been addressed in line 210 – 211.

Comment:

216-217: the concept of a transmission blocking vaccine (TBV) is explained as ‘TBV works by halting new infections in transmission-216 competent insects, by focusing on specific arthropod proteins’, which would benefit from a practical example (i.e. a specific protein that is blocked in one of the TBV approaches).

Response:

This has been addressed in line 254-266.

Comment:

Supplementary table: this is somewhat incomplete, it would be good to show the respective virus families, and then also order the agents according to the virus family (instead of alphabetically). How were the references selected that are given here?

Response:

The respective families have been included and the reference used has been properly addressed.

Round 2

Reviewer 4 Report

My suggestions have mostly been addressed by the authors. The only open point is still line 55-56: "infected arthropods rarely show disease manifestations, despite the existence of viral infection in the arthropod for a long time, up to months or years". I still think this is misleading and needs to be reworded 

Author Response

Dear Reviewer, 

Thank you for your insightful comment on out manuscript and we have addressed the manuscript as suggested in lines 54-57 of the revised draft.

Best regards

Authors
